# Dynamics of Microbial Community during the Co-Composting of Swine and Poultry Manure with Spent Mushroom Substrates at an Industrial Scale

**DOI:** 10.3390/microorganisms10102064

**Published:** 2022-10-19

**Authors:** Wan-Rou Lin, Han-Yun Li, Lei-Chen Lin, Sung-Yuan Hsieh

**Affiliations:** 1Bioresource Collection and Research Center (BCRC), Food Industry Research and Development Institute, Hsinchu 30062, Taiwan; 2Department of Forestry and Natural Resources, National Chiayi University, Chiayi 60004, Taiwan

**Keywords:** compost maturity, livestock and poultry manure, fungal community, bacterial community

## Abstract

Spent mushroom substrates (SMSs) can be developed as a biofertilizer through composting. Here, we investigated the dynamics of bacterial and fungal communities during commercial composting and the effect of swine and poultry manure on their communities through MiSeq pyrosequencing. *Weissella paramesenteroides* and *Lactobacillus helveticus* were dominant bacterial species in the composts with soy waste (SMS-SW), whereas Thermotogaceae sp. and *Ureibacillus* sp. were dominant in the composts with swine and poultry manure (SMS-PM). For the fungal community, *Flammulina velutipes* was dominant in SMS-SW, whereas *Trichosporon asahii*, *Candida catenulate*, *Aspergillus fumigatus*, and *Candida tropicalis* were dominant in SMS-PM. The addition of manure affected the bacterial community significantly. Redundancy analysis indicated that bacterial communities were affected by temperature, potassium, and potassium oxide and fungal communities by temperature, Kjeldahl nitrogen, organic matter, and ammonium nitrogen. Our findings can guide future research on composting microbiology.

## 1. Introduction

Mushrooms are crucial crops in Taiwan and other countries, and their total production in 2016 generated a revenue of approximately $430 million. Most cultivated mushrooms are grown on substrates comprising rice straw, rice bran, wheat bran, chicken manure, and sawdust or wood chip. Rice straw and sawdust derived from different types of trees are used as basic raw materials for cultivating edible mushrooms. After mushrooms are harvested, a considerable amount of the spent mushroom substrate (SMS) is left. According to Finney et al. [1], 1 kg of fresh mushrooms results in 5 kg of SMS (i.e., 2 kg dry weight). In the past, SMSs were considered as wastes and exerting adverse effects on the environments. These SMSs are rich in nutrients and can be applied as high-quality composts [2,3]) or soil conditioners in agriculture. Because of circular economy trends, SMSs are considered valuable resources for sustainable utilization.

Composting is a critical tool for the future of biological waste management [4]. Various microbes and their metabolites contribute to the composting process. The succession and dynamics of these microbes change with alterations in environmental parameters such as temperature, pH, and CO_2_ evolution rate [5]. Both culture-dependent and culture-independent methods have been used to describe changed in microbial composition during composting [5,6]. However, most studies have focused on the bacterial community in bench-scale composting reactors [7,8]. Few studies have focused on the dynamics of the fungal community and an industrial-scale composting system.

Commercial compost is prepared using a large-scale composting system involving various techniques such as in-vessel composting, open air-turned windrows, or in-vessel tunnel composting. In most cases, various starting materials of composting on an industrial scale were used, and few reports have focused on the dynamics of the microbial community on an industrial scale. In Taiwan, a rectangular agitated bed composting system in a building is used to process SMSs. The system was adopted in many areas because it encourages shorter composting periods.

Monitoring the composting process and changes in bacterial and fungal communities is essential to understand underlying mechanisms and obtain high-quality products. Here, we characterized the co-composting process of swine and poultry manure with SMSs in an industrial-scale composting system. Our specific objectives were (1) to study changes in bacterial and fungal communities during composting, (2) to investigate the physiochemical characteristics of composts during composting, and (3) to determine the relationship between the microbial community and compost characteristics.

## 2. Materials and Methods

### 2.1. Composting Materials and Processing

The experiments including one treatment and one control, and each group was conducted in two windrow tests. The treatment group (SMS-PM) was composed of SMS and swine and poultry manure, while the control contained the same composition, without swine poultry manure (SMS-SW). The SMS was obtained from several commercial *Flammulina velutipes* operations located in Taiwan. SMS-PM comprised a mixture of SMS with swine and poultry manure at a 6:4 ratio (*v*/*v*), whereas SMS-SW included SMS and soy waste at a 6:4 ratio (*v*/*v*). The weights of both starting materials were 200 tons. Before the start of the co-composting process, the starting materials were mixed and homogenized via a large stirring machine and the initial moisture content was adjusted to approximately 50–55% by water spraying.

A rectangular agitated bed composting system was used for processing the SMS. This system combines controlled aeration with periodic turning. The composting occurred between walls that form long, narrow channels, referred to as beds. A rail or channel on the top of each wall supports and guides a compost-turning machine. A loader places raw materials at the front end of the bed. As the turning machine moves forward on the rails, it mixes the compost and discharges the compost behind itself. With each turn, the machine moves the compost at a set distance toward the end of the bed. The turning machines work in a manner similar to windrow turners, using rotating paddles or flails to agitate materials, break up clumps of particles, and maintain porosity. The machine includes a conveyor to move the compost and work automatically. In this study, the bed dimensions were 67 m × 3 m × 1.5 m. The turning frequency was turned every 2 to 3 days. The composting period was 30 days in summer with daily turning and 60 days in winter with turning every 2 days.

### 2.2. Sampling

Compost samples were collected at three sampling sites in each treatment. The sampling sites were located at the front (sample 1), middle (sample 2), and back end (sample 3) of the bed. At each sampling site, we cut the compost into piles and obtained six samples randomly from exposed edges. DNA was extracted from each sample. Six DNA samples were obtained from each sampling site. All the six DNA samples from one sampling site were mixed, leading to three combined compost DNA samples per treatment.

### 2.3. DNA Extraction and High-Throughput Sequencing

DNA was extracted from 0.25 g of each compost sample by using a ZymoBIOMICS DNA Miniprep Kit (Zymo Research, Irvine, CA, USA). The six DNA samples obtained from one sampling site were mixed for next-generation sequencing. Tri-Biotech (Taipei, Taiwan) was entrusted to conduct subsequent-generation sequencing, and the 16S rRNA V3-V4 sequence of bacteria and the internal transcribed spacer (ITS) region of fungi were analyzed. The V3-V4 regions of the bacterial 16S rRNA gene were amplified using primer 341F (5′-CCT ACG GGN GGC WGC AG-3′) and 805R (5′-GAC TAC HVG GGT ATC TAA TCC-3′) [9] with different barcodes for the V3-V4 region of the 16S rRNA gene [10]. The ITS regions of the fungal rRNA gene were analyzed using primer ITS1F (5′-CTT GGT CAT TTA GAG GAA GTA A-3′) and ITS2 (5′-GCT GCG TTC TTC ATC GAT GC-3′) [11,12]. The 5′ end of 341F and ITS1F was barcoded with 8-bp error correcting barcodes [10] to enable sample multiplexing.

Each PCR was performed in a 25-μL reaction volume containing 1× HiFi Fidelity reaction buffer (Kapa Biosystems, Wilmington, MA, USA), 0.3 μM forward primer, 0.3 μM reverse primer, 1 μM dNTP mix (Kapa Biosystems), 0.5 U HiFi DNA polymerase (Kapa Biosystems), 2 ng DNA, and nuclease-free water. The amplification protocol comprised one denaturation step at 95 °C for 3 min, 25 cycles of denaturation at 98 °C for 20 s, annealing at 57.5 °C for 20 s, extension at 72 °C for 20 s, and a final extension at 72 °C for 3 min. A template-free reaction was used as the control. PCR products were separated in a 2.0% agarose gel (*w*/*v*) and purified using a QIAquick Gel Extraction Kit (Qiagen, New Delhi, India), according to the manufacturer’s instructions. The purified DNA was quantified using a Qubit dsDNA BR Assay Kit (Invitrogen, Waltham, MA, USA) in Qubit 2.0 Fluorometer (Invitrogen), and the individual samples were pooled in equimolar proportions. The final DNA pool was sent to the NGS facility at Tri-I Biotech (Taipei, Taiwan) for paired-end Illumina MiSeq sequencing.

The quality control of the raw sequencing data was performed using the quality control tool of the CLC Genomics Workbench. We adopted stringent conditions to process data and finally generated effective reads by modifying the MiSeq SOP [13,14]. All the effective reads obtained from all samples were clustered into operational taxonomic units (OTUs) at 97% sequence similarity by using USEARCH. Next, we classified OTUs based on reads taxonomy. If ≥51% reads in an OTU belonged to the same taxon (e.g., species), it was chosen as the taxonomic classification of the OTU. If <51% reads in an OTU belonged to the same taxon, the calculation was replicated at a higher taxonomic level (e.g., genus). The representative sequences of OTUs were taxonomically annotated using the SILVA database.

### 2.4. Physiochemical Analysis

Six compost samples from each sampling site were combined into one mixed compost sample. For each mixed compost sample, the physiochemical parameters were analyzed. Temperature was measured using a compost thermometer. The pH of the compost (1:10 *w/v* waste: water extract) was analyzed following the method reported by Kalamdhad et al. [15]. The moisture content was determined by drying the samples at 60 °C for 24 h. The electrical conductivity of the samples was analyzed using an electrical conductivity meter following Agriculture Fertilizer Standards (AFS) 2905-1. The total Kjeldahl nitrogen (TKN) content was analyzed using the Kjeldahl method [16]. The contents of ammonium nitrogen and total potassium were detected per AFS1111-1 and AFS2130-2, respectively. Volatile solids were determined using the loss ignition method (on a dry mass basis) at 550 °C for 2 h. The total organic carbon (TOC) content was calculated from volatile solids. The total phosphorus (TP; acid digest) content was examined using the stannous chloride method [17]. The potassium content was determined through atomic absorption, and the phosphorus content was determined colorimetrically following [18].

### 2.5. Statistical Analyses

The relative abundance of microbes in each sample was recorded as a binary matrix, which was used to calculate the Bray–Curtis similarity index [19] and construct nonparametric multidimensional scaling (MDS) plots using Primer 6 software (version 6.1.15; Primer-E, Plymouth, UK). The significance of differences found in the OTU-based structures of bacterial and fungal communities between SMS-PM and SMS-SW was assessed by performing the permutational multivariate analysis of variance (PERMANOVA) and analysis of similarity (ANOSIM) with Bray-Curtis similarities using Primer 6. Similarity percentage (SIMPER) analysis was performed to determine microbial OTUs that were primarily responsible for the observed differences by using Primer 6. Redundancy analysis [20], as implemented in the ‘‘Vegan’’ package for R software [21], was used to analyze relationships between physiochemical parameters and microbial profiles.

## 3. Results and Discussion

### 3.1. Physiochemical Characteristics of Composts

The physiochemical characteristics of the six samples are listed in Table 1. The temperature increased from 38.33 °C in the front to 43.50 °C in the middle and decreased from 43.50 °C to 39.33 °C in the back end in SMS-SW treatment. Similar trend also found in SMS-PW treatment. The moisture content decreased from the front to the back end in both the treatments. Microorganisms strongly decomposed the organic matters in the composts, which released heat and improved temperature. The high temperature not only promoted decomposition of the organic matters, but also accelerated water volatilization [22]. The temperature in the SMS-PM was higher than that in the SMS-SM. The total organic carbon content ranged from 93.40% to 96.83% in the SMS-SW samples and from 85.95% to 92.35% in the SMS-PM samples. The total organic carbon content in the SMS-SW samples was higher than that in the SMS-PW samples. This may because the addition of animal manure provides additional microorganisms whose respiration and activity can cause heat accumulation [23].

The total nitrogen content ranged from 28.70 to 82.85 mg/L in the SMS-SW samples and from 54.82 to 87.26 mg/L in the SMS-PM samples, whereas the total potassium content was approximately 2.42–2.49 mg/L and 2.93–3.99 mg/L, respectively. Nitrogen and potassium are essential nutrients for plant growth. Higher the total nitrogen and total potassium contents were observed in SMS-PW samples (Table 1). This indicated that the addition of manure to the SMS-PM compost demonstrated an increasing trend in the contents of total nitrogen and total potassium. Co-composting of swine and poultry manure with spent mushroom substrates could improve compost quality by increasing the nutritional content [24].

### 3.2. Bacterial and Fungal Diversity

The composition of bacterial 16S rRNA and fungal ITS sequences was determined using the illumina platform. In total, six DNA samples were amplified and sequenced. For bacteria, the effective reads generated by high-throughput sequencing were 43,920 (SMS-SW1), 42,365 (SMS-SW2), 43,295 (SMS-SW3), 41,682 (SMS-PM1), 42,226 (SMS-PM2), 41,616(SMS-PM3). In total, effective tags generated by high-throughput sequencing were aggregated at 97% sequence similarity, yielding 45,021 bacterial OTUs. These OTUs comprised 953 species, 772 genera belonging to 244 families.

Firmicutes, Thermotogae, Proteobacteria, Actinobacteria, and Bacteroidetes were the main bacterial phyla in all the samples. Firmicutes (66.37%) was the most dominant bacteria in all the samples, followed by Thermotogae (18.29%), Proteobacteria (7.29%), Actinobacteria (4.54%), and Bacteroidetes (3.25%). The communities at the genus level were considerably different between the two treatments. The bacterial compositions at the genus level are illustrated in Figure 1. *Weissella* and *Lactobacillus* were the dominant genera in SMS-SW. The relative abundance of *Weissella* was 57.06%, 52.95%, and 2.59% in SMS-SW1 (front), SMS-SW2 (middle), and SMS-SW3 (back), which decreased from the front site to the back site (Figure 1). The relative abundance of *Lactobacillus* increased from the front site (9.48%) to the back site (61.30%).

In this study, the relative abundance of *Weissella* spp. decreased from SMS-SW1 to SMS-SW3. *Weissella* was dominant in the early stages of SMS-SW composting but decreased in the late stage. *Weissella* is a Gram-positive lactic acid bacterium [25]. *Weissella* spp. have been found on the skin and in the milk and feces of animals; in the saliva, breast milk, feces, and vagina of humans; and in plants and vegetables [25,26]. *Weissella* were also dominant in the silage fermentation with soybean [27] and soy sauce fermentation [28]. The additional of soy waste might provide the bacterial inoculum. Tran et al. [29] found that *Weissella* inhibited vigorous organic matter degradation by accumulating acetic acid during composting process [2]. The decrease of *Weissella* in this study suggested the composting process was accelerated in the late stage. In this study, the relative abundance of *Lactobacillus* spp. increased from SMS-SW1 to SMS-SW3. *Lactobacillus* spp. could produce lactic acid and bacteriocins against pathogens [30]. The increase of their population in the late stage could increase the safety and quality of final products [31].

Themotogaceae S1 (26.98%) and *Ureibacillus* (11.88%) were the dominant genera in SMS-PW (Figure 1). Members of the Thermotogaceae family are moderately thermophilic to hyperthermophilic [32]. They degraded and utilized a wide range of simple and complex carbohydrates [33]. *Ureibacillus* comprises thermophilic, aerobic, endospore-forming bacteria. *Ureibacillus* spp. have been isolated from air [34], soil [35], and composts [36]. Most *Ureibacillus* species can grow at 35–65 °C, with the optimal growth temperature being 50–60 °C [36]. During composting, the temperature rises to 50–60 °C in SMS-PW treatment. Thus, hot compost represents a favorable habitat for these thermophilic isolates. Wan et al. [37] found that inoculation with a group of microorganisms consisting of *Bacillus*, *Ureibacillus*, *Geobacillus* and *Paracoccus* prolonged the thermophilic stage in composting, increasing the temperature with maximum temperature 68 °C. They suggest that inoculation with microorganisms were helpful in facilitating the process of composting.

For a detailed analysis of the bacterial community, 77 dominant species (the abundance >1% at any sample) were selected for MDS, PERMANOVA, and SIMPER analyses. The species-level MDS plot illustrates that bacterial communities in SMS-PW and SMS-SW were distinct (Figure 2). Our findings indicated that the addition of swine and poultry manure affected the bacterial community. However, the PERMANOVA test show the difference was not significant (*p* = 0.1). This might be caused by the limited sample sizes.

The SIMPER test indicated that Thermotogaceae sp., *W. paramesenteroides*, *Ureibacillus* sp., *Lactobacillus helveticus*, and *W. cibaria* were the main contributors to variations between bacterial communities in both the treatments, contributing 21.38%, 13.61%, 9.14%, 5.97%, and 5.06% dissimilarity, respectively (Table 2). *Weissella paramesenteroides*, *L. helveticus*, and *W. cibaria* were dominant in the SMS-SW samples, whereas Thermotogaceae sp. and *Ureibacillus* sp. were dominant in the SMS-PM samples (Table 2). Most strains of *W.*
*paramesenteroides* can grow at 10–37 °C, with the optimal temperature being 20–30 °C [25]. However, the thermophilic *W. paramesenteroides* can survive at temperatures as high as 55 °C in thermophilic composts [38]. *W. paramesenteroides* dominated in the SMS-SW samples where the temperature was 38–43 °C.

Basidiomycota (62.48%) and Ascomycota (35.11%) were the main fungi in all the samples. The fungal compositions at the genus level are illustrated in Figure 3. *Flammulina* (16.45%), *Candida* (14.44%), *Trichosporon* (10.82%), and *Thermomyces* (9.05%) were the dominant genera in SMS-SW, whereas *Thermomyces* (20.95%), *Trichosporon* (14.48%), and *Candida* (12.73%) were the dominant genera in SMS-PW (Figure 3).

For fungi, the effective reads generated by high-throughput sequencing were 41,353 (SMS-SW1), 42,726 (SMS-SW2), 39,068 (SMS-SW3), 39,959 (SMS-PM1), 38,566 (SMS-PM2), 40,821 (SMS-PM3). In total, effective tags generated by high-throughput sequencing were aggregated at 97% sequence similarity, yielding 741 fungal OTUs. These OTUs comprised 290 species, 176 genera belonging to 107 families. For a detailed analysis of the fungal community, 35 dominant species (the abundance >1% at any sample) were selected for MDS, PERMANOVA, and SIMPER analyses. The species-level MDS plot illustrates that fungal communities in SMS-PW and SMS-SW were non-significantly distinct (*p* = 0.4; Figure 4). The SIMPER test indicated that *Flammulina velutipes*, *Trichosporon asahii*, *Candida catenulate*, *Aspergillus fumigatus*, and *Candida tropicalis* were the main contributors to variations between fungal communities in both the treatments, contributing to 21.52%, 16.75%, 12.9%, 8.95%, and 7.47% dissimilarity, respectively (Table 2). *Flammulina velutipes* was dominant in the SMS-SW samples, whereas *Trichosporon asahii*, *Candida catenulate*, *Aspergillus fumigatus*, and *Candida tropicalis* were dominant in the SMS-PM samples (Table 2).

A shift in fungal taxa during composting was observed. *Thermomyces lanuginosus* was present in both the treated samples. The relative abundance was 2.69%, 2.25%, and 37.78% in three sampling sites in SMS-SW and 42.27%, 55.43%, and 1.26% in SMS-PM, respectively. *Thermomyces lanuginosus* was reported to be dominant in mushroom compost [23,39] and can degrade hemicellulose during composting [40]. The relative abundance of *Thermomyces lanuginosus* suggested that the degradation of hemicellulose appeared in the late period of SMS-SW compost and the early period of SMS-PM. Additionally, *T. lanuginosus* was abundant in the compost at a higher temperature (>55 °C). In summary, *T. lanuginosus* was widely distributed in mushroom composts, with higher abundance under higher temperatures.

The SMS was obtained from several commercial *F. velutipes* operations. *F. velutipes* was dominant in the SMS-SW samples. The relative abundance of *F. velutipes* decreased, indicating that their biomass was degraded during composting. *Candida tropicalis* and *Trichosporon asahii* were present in both the treated samples and were more abundant in SMS-PM. Two fungal species were opportunistic pathogens. Therefore, compost operators should limit the exposure and spread of airborne fungal spores and colonies during composting.

### 3.3. Correlations between Microbiota and Physiochemical Parameters during Composting

To determine physiochemical parameters that best explained the variation in microbial communities, redundancy analysis was performed to predict principal coordinates (Figure 5 and Figure 6) by using a linear combination of several physiochemical parameters (Table 1). These physiochemical factors, namely temperature; pH; moisture content; electrical conductivity value; TKN; and ammonium nitrogen, total potassium, TOC, and TP contents, accounted for 61.8% (Figure 5) and 36.55% (Figure 6) of the variation in bacterial and fungal communities, respectively. Relative abundance of Thermotogaceae_S1_unclassified and *Ureibacillus* sp. was increased with the increase of temperature, whiles some taxa dominant in SMS-SW, such as *Weissella cibaria* and *W. paramesenteroides* show the reverse trend. Total potassium (P) also raised with increased relative abundance of Thermotogaceae_S1_unclassified.

## 4. Conclusions

In the present study, shifts in microbial community and physiochemical characteristics were observed during composting through an industrial composting system. *Weissella paramesenteroides* and *Lactobacillus helveticus* were dominant bacterial species in the composts with soy waste (SMS-SW), whereas Thermotogaceae sp. and *Ureibacillus* sp. were dominant in the composts with swine and poultry manure (SMS-PM). For the fungal community, *Flammulina velutipes* was dominant in SMS-SW, whereas *Trichosporon asahii*, *Candida catenulate*, *Aspergillus fumigatus*, and *Candida tropicalis* were dominant in SMS-PM. Bacterial and fungal communities were changed by the addition of swine and poultry manure. Changes in physiochemical parameters could explain the variation in microbial communities. Bacterial communities were affected by temperature, potassium, and potassium oxide, whereas fungal communities were affected by temperature, Kjeldahl nitrogen, organic matter, and ammonium nitrogen. The data and analyses provided a rich source for additional investigations of composting microbiology.

## Figures and Tables

**Figure 1 microorganisms-10-02064-f001:**
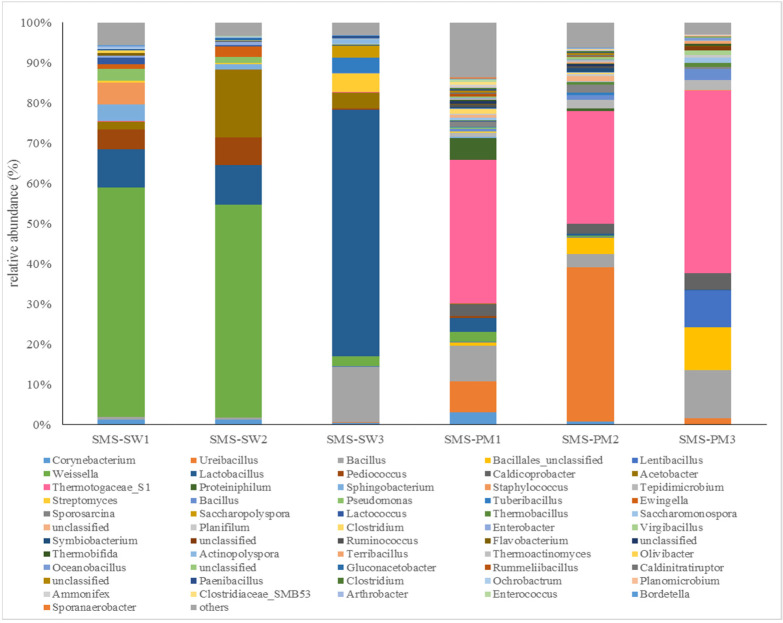
Dynamic changes in the bacterial community at the genus level from spent mushroom substrate with soy wastes (SMS-SW) and with swine and poultry manure (SMS-PW). The relative abundance of genus was >0.1% in at least one sample. Genera with relative abundance <0.1% were grouped as “others”.

**Figure 2 microorganisms-10-02064-f002:**
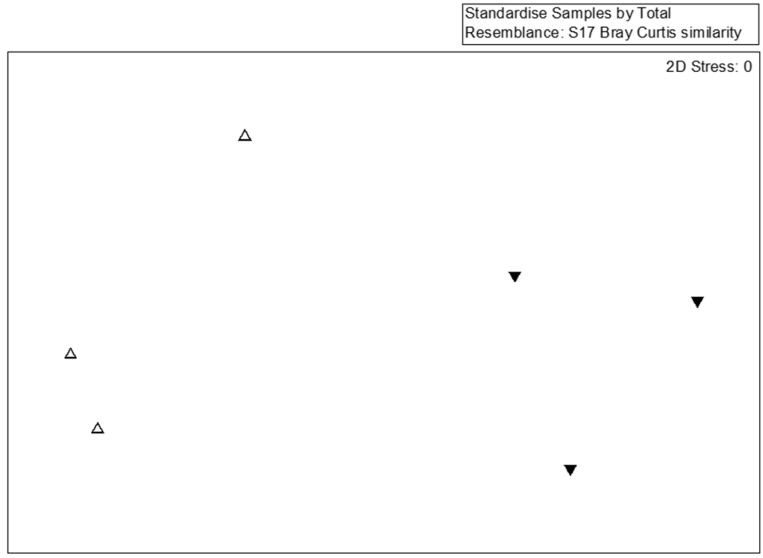
Multidimensional scaling of the bacterial community at the species lever from six compost samples. Bacterial taxa with relative abundances >0.1% in at least one sample were considered. Symbols indicate different treatments: triangle, spent mushroom substrate with soy wastes; inverted triangle, spent mushroom substrate with swine and poultry manure.

**Figure 3 microorganisms-10-02064-f003:**
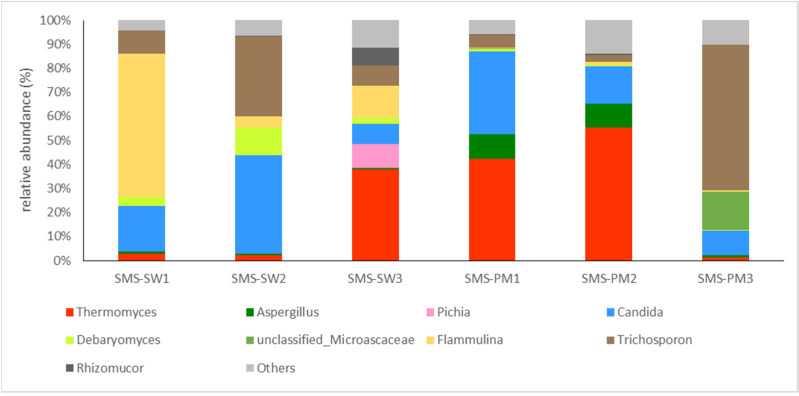
Dynamic changes of the fungal community at the genus level from spent mushroom substrate with soy wastes (SMS-SW) and with swine and poultry manure (SMS-PW). The relative abundance of genus was >0.1% in at least one sample. Genera with relative abundance <0.1% were grouped as “others”.

**Figure 4 microorganisms-10-02064-f004:**
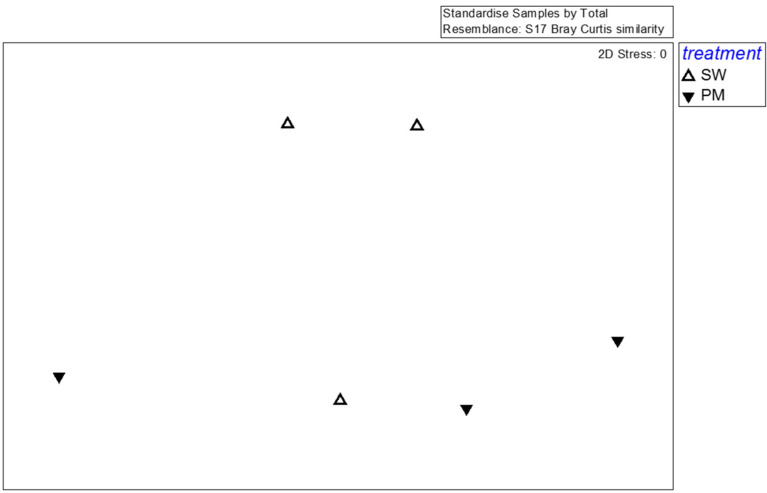
Multidimensional scaling of the fungal community at the species level from six compost samples. Fungal taxa with relative abundance >0.1% in at least one sample were considered. Symbols indicate different treatments: triangle, spent mushroom substrate with soy wastes; inverted triangle, spent mushroom substrate with swine and poultry manure.

**Figure 5 microorganisms-10-02064-f005:**
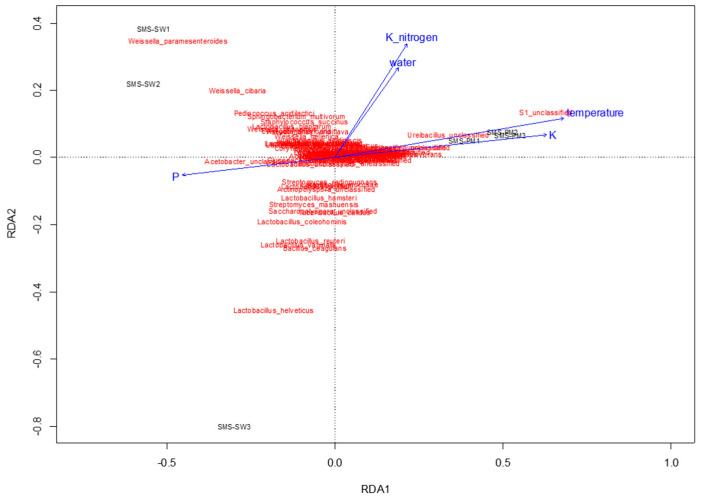
Redundancy analysis assessing the relationship between environmental factors (blue arrows) and bacterial communities (red). The arrows indicate the size and direction of the coefficients of physiochemical variables in the linear model.

**Figure 6 microorganisms-10-02064-f006:**
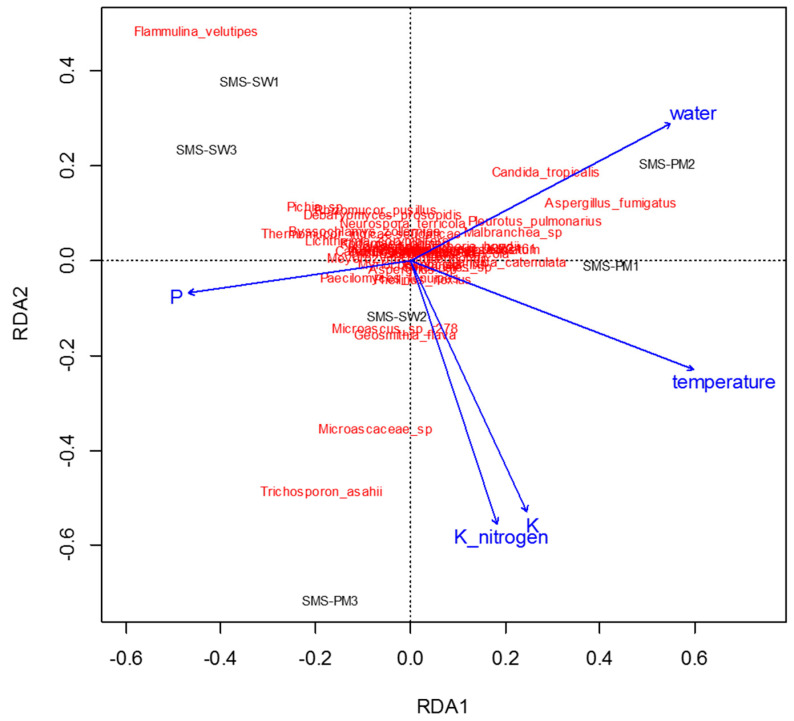
Redundancy analysis assessing the relationship between environmental factors (blue arrows) and fungal communities (red). The arrows indicate the size and direction of the coefficients of physiochemical variables in the linear model.

**Table 1 microorganisms-10-02064-t001:** Characteristics of physical-chemical during composting.

	Temperature (°C)	Moisture Content (%)	Total Kjeldahl Nitrogen (mg/L)	Total Phosphorus (mg/L)	Total Potassium (mg/L)	Total Organic Carbon (%)	Ammonium Nitrogen (%)	pH	EC (dS/m)
SMS-SW1	38.33	48.15	46.59	28,802.24	2.42	96.83	0.01	6.30	2.11
SMS-SW2	43.50	44.38	82.85	28,969.66	2.48	95.13	0.01	6.23	2.22
SMS-SW3	39.33	38.30	28.70	28,467.42	2.49	93.40	0.10	5.93	3.04
SMS-PM1	56.67	56.33	64.20	27,714.06	3.57	85.95	0.20	5.90	4.86
SMS-PM2	64.50	52.63	54.82	14,823.30	2.93	87.73	0.20	6.82	2.91
SMS-PM3	55.33	37.73	87.26	26,123.64	3.99	92.35	0.01	6.01	4.79

**Table 2 microorganisms-10-02064-t002:** Similarity percentage (SIMPER) analysis showing the five most influential species that contribute to the difference in the bacterial or fungal community structure between the spent mushroom substrate with addition of soy wastes (SMS-SW) and swine and poultry manure (SMS-PM).

	Species	Average of Relative Abundance (%)		Contribution (%)	Cumulative (%)
		SW	PM		
Bacteria	*Thermotogaceae* sp.	0.06	41	21.38	21.38
	*Weissella paramesenteroides*	26.75	0.9	13.61	34.99
	*Ureibacillus* sp.	0.08	17.58	9.14	44.13
	*Lactobacillus helveticus*	11.57	0.18	5.97	50.1
	*Weissella cibaria*	9.84	0.16	5.06	55.15

Fungi	*Flammulina velutipes*	29.97	1.44	21.52	21.52
	*Trichosporon asahii*	19.18	26.41	16.75	38.27
	*Candida catenulata*	18.15	25.58	12.9	51.17
	*Aspergillus fumigatus*	0.65	12.34	8.95	60.13
	*Candida tropicalis*	6.56	11.5	7.47	67.6

## Data Availability

Not applicable.

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
