# Peer review of "Dynamics of Microbial Community during the Co-Composting of Swine and Poultry Manure with Spent Mushroom Substrates at an Industrial Scale"

_microorganisms, 2022, doi:10.3390/microorganisms10102064_

Round 1

Reviewer 1 Report

The authors investigated the bacterial and fungal communities of commercial composting and the effect of swine and poultry manure on their communities, which help to understand underlying mechanisms and obtain high-quality products.

1、 How to understand the word "dynamics" in the title and main text? Because the compost samples were collected at 3 sites from 2 groups only at ONE point in time.

2、 Have you detected the microbial community of animal manure? In other words, dose the addition of manure affected the bacterial/fungal community because of the nutrient substance or the additional microorganisms in the manure?

3、 It’s confused that “the TOC content decreased in the SMS-SW samples and increased in the SMS-PM samples (Line 164-165)”. It seems that the data displayed so far in Table 1 can only support the comparison of the two groups but not the variation trend in each group.

4、 After the matched samples t- test, the P values of total nitrogen and total potassium are 0.59 and 0.08, respectively. It is hard to draw the conclusion that “a significant increase in the contents of total nitrogen and total potassium” rigorously (Line 173-174).

5、 Probably the biggest flaw in the experimental design is the sample size. The 3 biological replicates of SMS-SW/SMS-PM groups came from 3 different sites, the excessive intra-group variations and limited sample size probably mask some significant differences between these 2 groups.

6、 In 3.4, the authors determined physiochemical parameters that explained the variation in microbial communities. Further data mining can be considered to determine the key species that help to improve compost quality.

Author Response

The authors investigated the bacterial and fungal communities of commercial composting and the effect of swine and poultry manure on their communities, which help to understand underlying mechanisms and obtain high-quality products.

How to understand the word "dynamics" in the title and main text? Because the compost samples were collected at 3 sites from 2 groups only at ONE point in time.

Response 1: In this study, the rectangular agitated bed system was used for processing the SMS. The system has long, narrow beds in which to compost and an automated turner for periodic turning. The turner is supported on rails that are mounted on either side of the bed for its whole length. As the turner moves along the bed, the compost is turned and moved a set distance until it is ejected at the end of the bed. In the beginning of composting, all substrates were in the front site. After turner moves, the compost is turned and moved a set distance. Different sampling sites represented different time point.

Have you detected the microbial community of animal manure? In other words, does the addition of manure affected the bacterial/fungal community because of the nutrient substance or the additional microorganisms in the manure?

Response 2: Animal manures is one the common raw materials used in composting. Wang et al. (2018) found that Firmicutes often dominate in pig manure compost whiles Proteobacteria are more abundant in cattle manure. Such variations could be driven by the composting process, the nutrient substance or the additional microorganisms in the manure (Hussein et al., 2017; Chen et al., 2019). We did not analyze the microbial community of animal manure in this study. However, the high relative abundance of Ureibacillus spp. (7.65%) and Thermotogaceae_S1 (35.82%) in the beginning of SMS-PM (front site) composting suggested that these microbiota was introduced through the addition of pig manure.

It’s confused that “the TOC content decreased in the SMS-SW samples and increased in the SMS-PM samples (Line 164-165)”. It seems that the data displayed so far in Table 1 can only support the comparison of the two groups but not the variation trend in each group.

Response 3: We revised the sentence as follow. The total organic carbon content ranged from 93.40% to 96.83% in the SMS-SW samples and from 85.95% to 92.35% in the SMS-PM samples. The total organic carbon content in the SMS-SW samples was higher than that in the SMS-PW samples.

After the matched samples t- test, the P values of total nitrogen and total potassium are 0.59 and 0.08, respectively. It is hard to draw the conclusion that “a significant increase in the contents of total nitrogen and total potassium” rigorously (Line 173-174).

Response 4: Yes. We agreed that it’s not rigorous to draw the conclusion that a significant increase in the contents of total nitrogen and total potassium without statistical test. Therefore, we revised the sentence as follow. This indicated that the addition of manure to the SMS-PM compost demonstrated an increase trend in the contents of total nitrogen and total potassium.

Probably the biggest flaw in the experimental design is the sample size. The 3 biological replicates of SMS-SW/SMS-PM groups came from 3 different sites, the excessive intra-group variations and limited sample size probably mask some significant differences between these 2 groups.

Response 5: The bacterial/fungal community between SMS-PW and SMS-SW was not significantly differet. This might be caused by the limitaion of sample size. More samples within one treatment will be helpful. In spite of the fact, the shift in fungal and bacterial taxa between two groups was also observed.

In 3.4, the authors determined physiochemical parameters that explained the variation in microbial communities. Further data mining can be considered to determine the key species that help to improve compost quality.

Response 6: Thank you for the suggestions. We added some discussion as follow. Relative abundance of Thermotogaceae_S1_unclassified and Ureibacillus sp. was increased with the increase of temperature, whiles some taxa dominant in SMS-SW, such as Weissella cibaria and W. paramesenteroides show the reverse trend. Total potassium (P) also raised with increased relative abundance of Thermotogaceae_S1_unclassified.

Reviewer 2 Report

I found this article quite interesting and well written. However, I have some doubts about the presentation of the results.

 Lines 51-52: “In Taiwan, a rectangular agitated bed composting system in a building is used to process SMSs.”: Is this the only system adopted in Taiwan or is it the only system adopted in the present experiment?

Line 62: “The spent mushroom substrate was conducted in two windrow tests”: please rephrase.

Lines 88-89, lines 127-128: “Six compost samples from each sampling site were combined into one mixed compost sample”: although the experiment has three true experimental replicates (sampling sites), I would have preferred that even within one site the samples had been kept separate (at least for the physiochemical parameters). It is correct to keep the analyses of the three sampling sites separate. However, using a composite sample for each site, the experiment lacks replicates and this aspect should be discussed.

Figure 2, figure 4: please present graphs with at least indicated the titles of the axes!

Line 299-300: “The arrows indicate the size and direction of the coefficients of physiochemical 299 variables in the linear model (Figures 5 and 6)” this sentence should be placed in the figures, not in the results.

Figures 5 and 6: I think it is useless to report the names of the organisms in the figures, as they are (almost) all illegible. In the center of the figure, only a single indistinguishable red spot can be appreciated.

Conclusion: I find the conclusions a bit too short. I believe the conclusion could be developed better and in the light of the bibliography cited in the introduction

Author Response

Response to Reviewer 2 Comments

I found this article quite interesting and well written. However, I have some doubts about the presentation of the results.

Lines 51-52: “In Taiwan, a rectangular agitated bed composting system in a building is used to process SMSs.”: Is this the only system adopted in Taiwan or is it the only system adopted in the present experiment?

Response 1: The rectangular agitated bed system uses long, narrow beds in which to compost and an automated turner for periodic turning. The turner is supported on rails that are mounted on either side of the bed for its whole length. As the turner moves along the bed, the compost is turned and moved a set distance until it is ejected at the end of the bed. The system was adopted in many areas because the system encourage shorter composting periods.

Line 62: “The spent mushroom substrate was conducted in two windrow tests”: please rephrase.

Response 2: We revised it as follow. The experiments including one treatment and one control was conducted in two windrow tests, respectively: an experimental test with the addition of swine and poultry manure (SMS-PM) and one control test (SMS-SW).

Lines 88-89, lines 127-128: “Six compost samples from each sampling site were combined into one mixed compost sample”: although the experiment has three true experimental replicates (sampling sites), I would have preferred that even within one site the samples had been kept separate (at least for the physiochemical parameters). It is correct to keep the analyses of the three sampling sites separate. However, using a composite sample for each site, the experiment lacks replicates and this aspect should be discussed.

Response 3: Thank you. In the pre-test for the physicochemical parameter, the replicates within one site show similar data. Therefore, six compost samples from each sampling site were combined into one mixed compost sample. We agree that the replicates in each sample may provide more robust result.

Figure 2, figure 4: please present graphs with at least indicated the titles of the axes!

Response 4: Figure 2 and Figure 4 showed the multidimensional scaling of the bacterial and fungal community, respectively. Multidimensional scaling (MDS) is an approach for graphically representing relationships between objects (e.g. compost samples) in multidimensional space. Dimension reduction via MDS is achieved by taking the original set of samples and calculating a dissimilarity measure for each pairwise comparison of samples. The samples are then usually represented graphically in two dimensions such that the distance between points on the plot approximates their multivariate dissimilarity as closely as possible. The two axes imply the variation in data along the two principal components.

Line 299-300: “The arrows indicate the size and direction of the coefficients of physicochemical 299 variables in the linear model (Figures 5 and 6)” this sentence should be placed in the figures, not in the results.

Response 5: Thank you. We revised it as suggestion.

Figures 5 and 6: I think it is useless to report the names of the organisms in the figures, as they are (almost) all illegible. In the center of the figure, only a single indistinguishable red spot can be appreciated.

Response 6: Thank you. The names of the organisms show some information. For example, relative abundance of Thermotogaceae_S1_unclassified and Ureibacillus sp. was increased with the increase of temperature, whiles some taxa dominant in SMS-SW, such as Weissella cibaria and W. paramesenteroides show the reverse trend. Therefore, we keep the names of the organisms in the figure and added some information in the revised manuscript.

Conclusion: I find the conclusions a bit too short. I believe the conclusion could be developed better and in the light of the bibliography cited in the introduction.

Response 7: Thank you. We revised it as follow. In the present study, shifts in microbial community and physicochemical characteristics were observed during composting through an industrial composting system. Weissella paramesenteroides and Lactobacillus helveticus were dominant bacterial species in the composts with soy waste (SMS-SW), whereas Thermotogaceae sp. and Ureibacillus sp. were dominant in the composts with swine and poultry manure (SMS-PM). For the fungal community, Flammulina velutipes was dominant in SMS-SW, whereas Trichosporon asahii, Candida catenulate, Aspergillus fumigatus, and Candida tropicalis were dominant in SMS-PM. Bacterial and fungal communities were affected by the addition of swine and poultry manure. Changes in physiochemical parameters could explain the variation in microbial communities. Bacterial communities were affected by temperature, potassium, and potassium oxide, whereas fungal communities were affected by temperature, Kjeldahl nitrogen, organic matter, and ammonium nitrogen. The data and analyses provided a rich source for additional investigations of composting microbiology.

Reviewer 3 Report

Dear authors,

The article By Lin et al. describes the composting of Spent mushroom substrates as biofertilizers is quite interesting to me and the methods and materials used to obtain the results are well described. The authors used soy waste or swine and poultry manure to compost the substrate using a rectangular agitated bed composting system. Microbial and physiochemical parameters were analyzed and correlations made.

However, my major concern is that the “process” of composting was not properly tracked as indicated in my comments (in the pdf file).

The Discussion of the results must be improved by providing more details (with specifics) that will help to make better sense of the seemingly good results.

I wish you the best of Luck^^

Author Response

The article By Lin et al. describes the composting of Spent mushroom substrates as biofertilizers is quite interesting to me and the methods and materials used to obtain the results are well described. The authors used soy waste or swine and poultry manure to compost the substrate using a rectangular agitated bed composting system. Microbial and physiochemical parameters were analyzed and correlations made. However, my major concern is that the “process” of composting was not properly tracked as indicated in my comments (in the pdf file). The Discussion of the results must be improved by providing more details (with specifics) that will help to make better sense of the seemingly good results.

Response: Thank you for the suggestion. We revised the manuscript as suggestion.

2.2 Sampling

I think the data obtained using this sampling criteria will not reflect the process but rather the compost state at that particular sampling time. I really think sampling at different times would have help. For instance, if the microbial community at the start, mid composting and final phase are different, that could mean a lot in terms of the treatments used. maybe the beneficial microbes increase, or the abundance of harmful microbes decline, or simply the succession of microbes depending on their metabolic lifestyles, heat tolerance etc. This is important because you would want to know the starting point, ....... to end point in order to define the physiochemical and microbial processes occuring in the compost. Im my opinion, the phrase "During Composting" in itself indicates a process rather than a "one point activity. "

Response: Thank you for the comments. In this study, we used different sampling sites represented different time point. The rectangular agitated bed system was used for processing the SMS. The system has long, narrow beds in which to compost and an automated turner for periodic turning. The turner is supported on rails that are mounted on either side of the bed for its whole length. As the turner moves along the bed, the compost is turned and moved a set distance until it is ejected at the end of the bed. In the beginning of composting, all substrates were in the front site. After turner moves, the compost is turned and moved a set distance.

Line 121-Line 123

“If ≥51% reads in an OTU belonged to the same taxon (e.g., species), it was chosen as the taxonomic classification of the OTU. If <51% reads in an OTU belonged to the same taxon, the calculation was replicated at a higher taxonomic level (e.g., genus).” Why 51%?

Response: We followed the methods from Yang et al. (2015). That 51% reads in an OTU belonged to the same taxon represented the dominant reads in this OTU belonged to one taxon or one species. Therefore, we used this species or taxon to name the OTU.

I don't get the essence of this table!

Because:

  1. the temperatures and all these other parameters will change along the composting process. So it wpuld have been more informative to show the trends in the composting process.
  2. It seems the samples were taken awhen the compsot was still at thermophilic phase. such temperature indicate active microbial activity. The compost has not attained maturation stage and can not be applied to field under such conditions as it would cause phytotoxicity. It would have been nice to show the values of the mature compost.

Since the nutrient content of the starting materials are not provided, how do you attribute the higher nutrient content to the composting process. maybe it just a reflection of the nutrients contained in starting materials.

These prameters like temp, nutrient content, pH, EC should be monitored along the process to help in predict the biological activities occuring in the compost.

  1. Maybe it would also help if you provide expected values and compare them with the measured values as a way of evaluating the product.

Response: Thank you. We added some discussion about the relative abundance of dominant species and temperature or other parameters. relative abundance of Thermotogaceae_S1_unclassified and Ureibacillus sp. was increased with the increase of temperature, whiles some taxa dominant in SMS-SW, such as Weissella cibaria and W. paramesenteroides show the reverse trend. Therefore, we keep the names of the organisms in the figure and added some information in the revised manuscript.

I appreciate the concept of relative abundance of microbial community in the compost. However, if we do not have an idea of the microbial density (population ~ probably estimated from the total reads after cleaning out the false reads), the relative abundance becomes less informative and may not necessarily give insight to the actual microbial activity occuring in the compost.

E.g., [hypothetically]: if Weissella spp. are inhibiting the composting process and are making up 50% of the total microbes in SW1 but only makes 5% in SW3. This may look like SW3 is composting at a better rate than SW1 but actually, this won't be the case is SW3 has substantial higher microbial population. Because 5% of a higher population could mean the same as 50% of a small population.

Response: The effective reads generated by high-throughput sequencing were ranged from 41,616 to 43,920. The numbers of effective reads among samples were similar. Therefore, the relative abundance of microbial community was comparable. We added the description of the total reads in the manuscript. In total, six DNA samples were amplified and sequenced. For bacteria, the effective reads generated by high-throughput sequencing were 43,920 (SMS-SW1), 42,365 (SMS-SW2), 43,295 (SMS-SW3), 41,682 (SMS-PM1), 42,226 (SMS-PM2), 41,616(SMS-PM3). In total, effective tags generated by high-throughput sequencing were aggregated at 97% sequence similarity, yielding 45,021 bacterial OTUs. These OTUs comprised 953 species, 772 genera belonging to 244 families. For fungi, the effective reads generated by high-throughput sequencing were 41,353 (SMS-SW1), 42,726 (SMS-SW2), 39,068 (SMS-SW3), 39,959 (SMS-PM1), 38,566 (SMS-PM2), 40,821 (SMS-PM3). In total, effective tags generated by high-throughput sequencing were aggregated at 97% sequence similarity, yielding 741 fungal OTUs. These OTUs comprised 290 species, 176 genera belonging to 107 families.

Also highlight metabolioc lifestyle of the microbes that show a higher correlation with a particular physiochemical state. Why is Uribacillus showing a better correlation temp, K N and H2O? oe Acetobacter with P? Why Lactobacillus has a negative correlation with moisture, N, temp and K? Based on the literature, could you provide light as to whether the optimizing such conditions could favour certain microbes that confer composting advantages? Or could the physiochemical condtions of the starting materials explain the differences in the microbial stracture? and how this could this inturn affect the composting process? Same for the fungal community stracture

Response: Relative abundance of Thermotogaceae_S1_unclassified and Ureibacillus sp. was increased with the increase of temperature, whiles some taxa dominant in SMS-SW, such as Weissella cibaria and W. paramesenteroides, show the reverse trend. Members of the Thermotogaceae family are moderately thermophilic to hyperthermophilic (Dahle et al. 2011). They degraded and utilized a wide range of simple and complex carbohydrates (Huber et al. 1986). Ureibacillus comprises thermophilic, aerobic, endospore-forming bacteria. Most Ureibacillus species can grow at 35–65 °C, with the optimal growth temperature being 50–60 °C (Kim et al. 2006). During composting, the temperature rises to 50–60 °C in SMS-PW treatment. Thus, hot compost represents a favourable habitat for these thermophilic isolates.

The conclusion is good but too short and less elaborate. How did swine and poultry affect bactera and fungal community. be specific, and what is the implication in relation to your objectives. How where the bacerial community affected by those physiochemical conditions, and what is the relevance in terms of composting composting process? Adding a sentence to eloborate on each of the key take away points could be good.

Response: Thank you. We revised it as follow. In the present study, shifts in microbial community and physiochemical characteristics were observed during composting through an industrial composting system. Weissella paramesenteroides and Lactobacillus helveticus were dominant bacterial species in the composts with soy waste (SMS-SW), whereas Thermotogaceae sp. and Ureibacillus sp. were dominant in the composts with swine and poultry manure (SMS-PM). For the fungal community, Flammulina velutipes was dominant in SMS-SW, whereas Trichosporon asahii, Candida catenulate, Aspergillus fumigatus, and Candida tropicalis were dominant in SMS-PM. Bacterial and fungal communities were affected by the addition of swine and poultry manure. Changes in physiochemical parameters could explain the variation in microbial communities. Bacterial communities were affected by temperature, potassium, and potassium oxide, whereas fungal communities were affected by temperature, Kjeldahl nitrogen, organic matter, and ammonium nitrogen. The data and analyses provided a rich source for additional investigations of composting microbiology.

Round 2

Reviewer 1 Report

accept

Author Response

Thank you.

Reviewer 3 Report

Dear Authors

I am ok with the response to my major concern about sampling time. To provide more clarity in regard, please provide the exact time when samples 1, 2, and 3 were taken. My understanding from your response is that sample 1 was taken at the beginning, 2 mid-way, and 3 towards the end. if this is true, please elaborate on it better in the M&M.
Anyways, your manuscript can be improved (including minor language errors).
Good Luck^^

Author Response

Dear Authors

I am ok with the response to my major concern about sampling time. To provide more clarity in regard, please provide the exact time when samples 1, 2, and 3 were taken. My understanding from your response is that sample 1 was taken at the beginning, 2 mid-way, and 3 towards the end. if this is true, please elaborate on it better in the M&M.
Anyways, your manuscript can be improved (including minor language errors).
Good Luck^^

Response: We revised the manuscript as suggestions. Thank you.